# Additional Mosapride to Proton Pump Inhibitor for Gastroesophageal Reflux Disease: A Meta-Analysis

**DOI:** 10.3390/jcm9092705

**Published:** 2020-08-21

**Authors:** Toshihiro Nishizawa, Kiyoto Mori, Shuntaro Yoshida, Hirotoshi Ebinuma, Osamu Toyoshima, Hidekazu Suzuki

**Affiliations:** 1Department of Gastroenterology and Hepatology, International University of Health and Welfare, Narita Hospital, Narita 286-8520, Japan; nisizawa@kf7.so-net.ne.jp (T.N.); kmrz2000@gmail.com (K.M.); ebinuma@me.com (H.E.); 2Gastroenterology, Toyoshima Endoscopy Clinic, Tokyo 157-0066, Japan; yoshidash-int@h.u-tokyo.ac.jp (S.Y.); t@ichou.com (O.T.); 3Department of Gastroenterology, Graduate School of Medicine, The University of Tokyo, Tokyo 113-0033, Japan; 4Division of Gastroenterology and Hepatology, Department of Internal Medicine, Tokai University School of Medicine, Isehara 259-1193, Japan

**Keywords:** mosapride, GERD, meta-analysis, PPI

## Abstract

Background and Aim: In gastroesophageal reflux disease (GERD), the additive effect of mosapride to a proton pump inhibitor (PPI) is still controversial. This meta-analysis integrated randomized controlled trials (RCTs) in which mosapride combined with a PPI was compared with a PPI alone in GERD treatment. Methods: RCTs were systematically searched with the PubMed, Cochrane library, Web of Science, and the Igaku-Chuo-Zasshi database. We combined the data from the RCTs with a random effects model, calculated the standardized mean difference (SMD) and pooled the risk difference (RD) with 95% confidence intervals (CIs). Results: We included nine RCTs in the present meta-analysis. In the mosapride combined with PPI group, the improvement of the symptom score was significantly greater than that in the PPI alone group without significant heterogeneity (SMD: −0.28, 95% CI: −0.45 to −0.12, *p* = 0.0007). In the mosapride combined with PPI group, the symptom score after treatment was significantly lower than that in the PPI alone group (SMD: −0.24, 95% CI: −0.42 to −0.06, *p* = 0.007). Conclusions: Mosapride combined with a PPI significantly improved the reflux symptom score compared with that of PPI alone.

## 1. Introduction

Gastroesophageal reflux disease (GERD) is often encountered in clinical practice. The characteristics are reflux symptoms such as acid regurgitation and heartburn [1,2,3,4]. The pathophysiology of GERD includes a disorder of motor function [5,6]. GERD often causes troublesome symptoms, and impairs quality of life (QOL) [7,8,9]. Proton pump inhibitors (PPIs) improve the QOL in GERD patients [10]. However, PPIs are sometimes ineffective against GERD.

An alternative approach to manage symptomatic GERD is to prevent acid reflux. There is a theory that prokinetic agents would be effective against GERD, because they could enhance lower oesophageal sphincter pressure, and improve oesophageal peristalsis, oesophageal acid clearance, and gastric emptying [11]. Mosapride citrate is a prokinetic drug which stimulates the 5-hydroxytryptamine 4 (5-HT4) receptor. Mosapride increases the release of acetylcholine from parasympathetic nerves and promotes gastrointestinal motility and gastric emptying. A randomized controlled trial (RCT) by Madan et al. showed that mosapride combined with a PPI exerted a more beneficial effect than PPI alone [12]. However, other RCTs reported a limited effect of mosapride [13,14,15]. In 2013, a systematic review concluded that mosapride combined with a PPI was not more effective than a PPI alone [16]. Since then, several RCTs have been conducted [17,18,19,20,21], and most RCTs showed that the improvement in GERD patients who received mosapride was not statistically significant, although mosapride seemed to offer a favorable result in these patients. The reason is that the sample size was not large enough to reach statistical significance. If the data from all RCTs are systematically integrated, statistically conclusive results may be achieved in the efficacy of mosapride. Therefore, we updated a meta-analysis of the RCTs evaluating the additional effect of mosapride to a PPI for GERD.

## 2. Methods

### 2.1. Search Strategy

The literature was systematically searched using the PubMed, Cochrane library, Web of Science, and the Igaku-Chuo-Zasshi database in Japan (up to May 2020) [22]. The search words in our systematic review were: (mosapride) AND (proton pump inhibitor) AND (reflux) AND (randomized). There was no limitation in language.

### 2.2. Inclusion and Exclusion Criteria

The inclusion criteria were: (1) study design: RCT; (2) participants of RCT: patients who had GERD; (3) intervention: mosapride combined with a PPI; (4) control: PPI; (5) outcome: therapeutic effect of mosapride for GERD. The exclusion criteria were: (1) non-use of PPIs by subjects in the control group; (2) meeting abstracts; (3) duplication; (4) review articles.

### 2.3. Outcome Measures

The primary outcome was improvement in the symptom scores in our meta-analysis. The secondary outcomes included the symptom scores after treatment, the response rate associated with the treatment, and the rate of adverse effects.

### 2.4. Data Extraction

We extracted the following data from the eligible RCTs: principal author, publishing year, country, details of patients (number, age, and sex), assessment method of reflux symptoms, the duration of therapy, drugs and doses for the treatment, and outcomes. The reflux symptom scores were different among the included studies. We converted the scores to a 0–10 scale. Two researchers (K.M. and T.N.) independently checked all articles for eligibility. A third reviewer (H.S.) resolved disagreements in consultation. We contacted the responding authors to clarify the details of the studies.

### 2.5. Assessment of Methodological Quality

We assessed the quality of the included literature with the risk-of-bias tool according to the Cochrane Handbook for Systematic Reviews of Interventions (version 5.1.0) [23]. Two reviewers (H.E. and O.T.) scrutinized all studies. The six items for RCT quality assessment were: (1) random sequence generation; (2) allocation concealment; (3) blinding of participants and outcome assessment; (4) assessment of incomplete outcome data; (5) completeness of outcome reporting; (6) other potential bias.

### 2.6. Statistical Analysis

We conducted statistical analysis with the Review Manager (RevMan; The Cochrane Collaboration, 2008; The Nordic Cochrane Centre, Copenhagen, Denmark) [24,25,26]. A random effects model and Mante–Haenszel method were used to calculate the risk difference (RD). Inverse variance for the continuous data was used to estimate the standardized mean difference (SMD) with a 95% confidence interval (CI). Heterogeneity between the studies was assessed with Cochran’s Q and I^2^ tests. A *p* value <0.1 was considered as significant heterogeneity because the power of the Q test is low. An I^2^ score ≥50% was considered as a moderate level of heterogeneity [27]. A sensitivity analysis was added to evaluate the stability of this meta-analysis. We divided all eligible trials into a non-erosive GERD group and an erosive esophagitis group, and subgroup analysis was also performed. Finally, publication bias was checked by funnel plot asymmetry. Egger’s regression test also examined funnel plot asymmetry [28,29,30,31].

## 3. Results

### 3.1. Search Results

Sixty-two citations were involved in the systematic review process (Figure 1). Among them, we excluded 50 studies according to the exclusion criteria (12 unrelated topics, 7 meeting abstracts, 25 duplications, 4 review articles, and 2 protocols for RCT). The remaining 12 studies were scrutinized, after which three more studies were rejected [32,33,34] because subjects in the control groups did not use PPIs. Finally, we included nine RCTs in our meta-analysis [12,13,14,15,17,18,19,20,21]. The details of the included RCTs are presented in Table 1.

### 3.2. Quality Assessment

Table 2 shows the risk of bias summary. In one RCT, allocation concealment was not described. In two RCTs, the patients and the outcome assessment were not blinded. In all RCTs, the incomplete outcomes were adequately assessed, and selective outcome reporting was avoided. All RCTs did not have other biases. In general, the quality of the RCTs was very good.

### 3.3. Meta-Analysis Results

#### 3.3.1. Improvement in Symptom Scores

A change in symptom score was described in six studies. In the mosapride combined with PPI group, the improvement of the symptom score was significantly greater than that in the PPI alone group (SMD: −0.28, 95% CI: −0.45 to −0.12, *p* = 0.0007, Figure 2A). Heterogeneity was not detected between RCTs (*p* = 0.70, I^2^ = 0%). The sensitivity analysis sequentially excluded one trial at a time, and did not change the meta-analysis results.

#### 3.3.2. Symptom Scores after Treatment

The symptom scores after treatment were described in seven studies. In the mosapride combined with PPI group, the symptom score after treatment was significantly lower than that in the PPI alone group (SMD: −0.24, 95% CI: −0.42 to −0.06, *p* = 0.007, Figure 2B). Heterogeneity was not detected between the seven studies (*p* = 0.36, I^2^ = 9%). The sensitivity analysis sequentially excluded one trial at a time, and did not change the meta-analysis results. The funnel plot was almost symmetrical (Figure 3). Egger’s regression test confirmed no evidence of substantial publication bias (*p* = 0.694).

#### 3.3.3. Response to Treatment

The symptom response rates were described in three studies. Miwa et al. defined the responders as those who scored less than one on the visual analogue scale (range of 0–10). Madan et al. defined the responders as those who reported a symptom score of ≤4 (range of 0–18). Cho et al. defined the responders as those who had symptoms decrease from a moderately severe or greater rating, to mild or absent. Due to limited numbers, RCTs that used different criteria were included in this meta-analysis. There was no statistically significant difference in symptom response between the PPI alone group and the mosapride combined with PPI group (pooled RD: −0.09, 95% CI: −0.20 to 0.02, *p* = 0.12, Figure 4A).

#### 3.3.4. Adverse Events

Adverse events were described in six studies. Adverse events were similar between the PPI alone group and the mosapride combined with PPI group (pooled RD: 0.00, 95% CI: −0.10 – 0.40, *p* = 0.84, Figure 4B).

#### 3.3.5. Subgroup Analysis

Two studies (Hsu et al. and Lee et al.) included patients with erosive esophagitis, and one study (Miwa et al.) included patients with non-erosive GERD. Madan et al. had subgroup analysis between non-erosive GERD and erosive esophagitis. In the study by Sirinawastien et al., the majority was non-erosive GERD (86.4%). Subgroup analysis between non-erosive GERD and erosive esophagitis was performed in the study by Sirinawastien et al. as a non-erosive GERD study, owing to the limited number of reports. Subgroup analysis indicated that mosapride combined with a PPI significantly improved the reflux symptom score compared with that of PPI alone for the erosive esophagitis group (*p* = 0.05), but no difference was found for the non-erosive GERD group (Figure 5A,B). Furthermore, mosapride combined with a PPI significantly enhanced the response rate compared with that of PPI alone for the erosive esophagitis group (*p* = 0.001), but no difference was found for the non-erosive GERD group (Figure 5C,D).

## 4. Discussion

Overall, we found that mosapride combined with a PPI significantly improved the reflux symptom score compared with that of PPI alone.

The previous systematic review by Liu et al. did not find any advantage of the addition of mosapride to PPIs in GERD patients [16]. The previous systematic review included four RCTs and analyzed the response rate. On the other hand, our meta-analysis included nine RCTs, and analyzed not only the response rate but also the symptom score. The response rates were described in only three studies, so did not reach significance in the previous systematic review or our meta-analysis. However, we have found that mosapride combined with a PPI significantly improved the symptom score.

5-HT4 receptor agonists include cisapride, mosapride, and tegaserod. Cisapride was on the market worldwide in the 1990s, and became popular as a prokinetic agent. In 2000, cisapride was removed from the global market because of serious cardiac adverse events [35]. Mosapride is a selective 5-HT4 receptor agonist, and has no effect on 5-HT1 and 5-HT2 receptors [16]. Mosapride does not affect potassium channels, as opposed to cisapride [36]. In several studies of healthy volunteers who received mosapride, no changes were detected in electrocardiogram results [37,38]. Mosapride is believed to be safe and without risk of cardiac adverse events. On the other hand, in 2007, tegaserod was removed from the market due to an increased risk of cardiovascular adverse events. In 2019, tegaserod was approved again for irritable bowel syndrome with constipation in women. However, the use is limited to women who are aged <65 years and who are at low risk for cardiovascular events [35].

Mosapride not only accelerates oesophageal acid clearance but may also improve the pharmacokinetics of PPIs. PPIs are unstable in gastric acid. The long retention of PPIs in the stomach could impair the acid-suppressive effect. Arai et al. showed that the pharmacokinetics of PPIs are improved by mosapride. The addition of mosapride significantly increased the area under the time–concentration curve (AUC), and the maximum plasma concentration [39]. Therefore, coadministration of mosapride could facilitate the rapid transit of PPIs into the intestine and exert better therapeutic effects.

In our subgroup analysis between non-erosive GERD and erosive esophagitis, mosapride combined with a PPI significantly improved the reflux symptom score and response rate compared with those of PPI alone for the erosive esophagitis group, but no difference was found for the non-erosive GERD group. The RCT by Madan et al. had subgroup analysis between non-erosive GERD and erosive esophagitis [12]. Mosapride combined with a PPI significantly improved the response rate compared with that of PPI alone for the erosive esophagitis group (*p* = 0.003), but no difference was found for the non-erosive GERD group. It is well known that PPIs work better in patients with erosive esophagitis than in those without erosions. Mosapride may also work better in patients with esophagitis than in those without erosions.

The cost of esomeprazole is 235 yen ($2.2) for 40 mg (daily dose). The cost of mosapride is 43 yen ($0.4) for 15 mg (daily dose). Mosapride combined with a PPI increases costs by 18%. We consider that this increase in cost would be acceptable for patients receiving PPIs with incomplete symptom control.

Our meta-analysis includes several limitations. Various methods for assessment of reflux symptoms might be considered a source of heterogeneity. Various doses of PPI might also be considered a source of heterogeneity. Furthermore, we did not perform subgroup analysis for the assessment methods and PPI doses, because of the limited number of eligible RCTs. There were no European RCTs because mosapride is not available in Europe. Response rates associated with the treatment did not reach statistical significance. Future studies are needed to clarify the efficacy of mosapride for GERD.

In conclusion, the addition of mosapride to a PPI improves reflux symptoms, especially for erosive esophagitis. Mosapride could be a promising option for patients receiving acid suppression agents with incomplete symptom control.

## Figures and Tables

**Figure 1 jcm-09-02705-f001:**
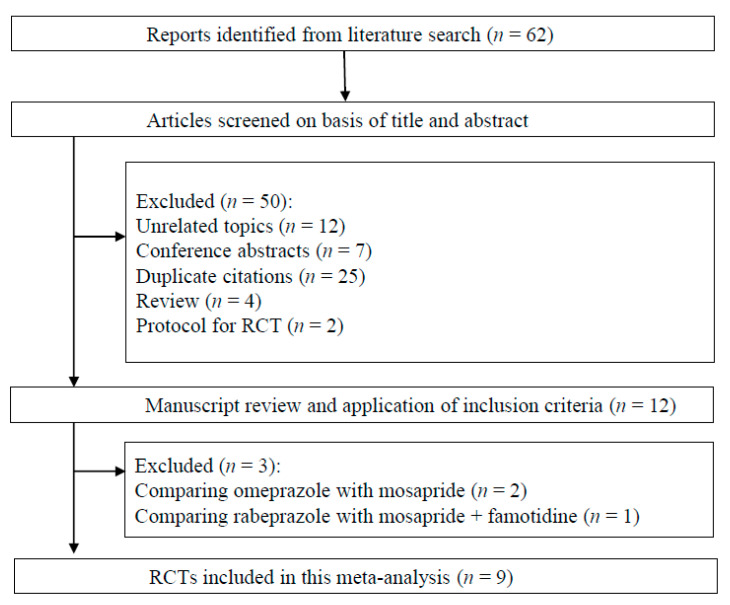
Flow diagram of the systematic literature search.

**Figure 2 jcm-09-02705-f002:**
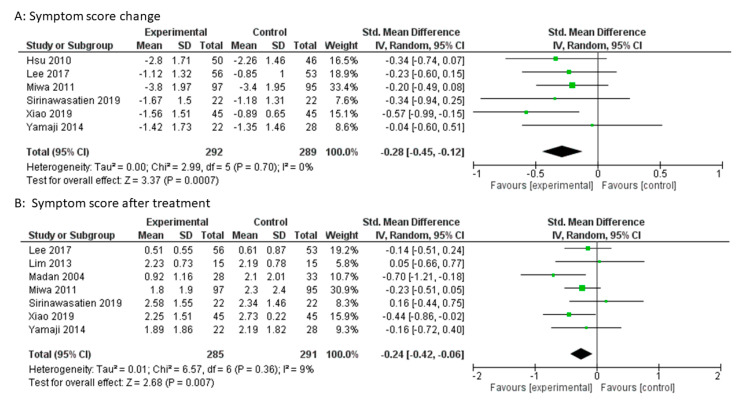
Forest plot of the standardized mean difference (SMD) with 95% confidence intervals (CI) for mosapride combined with PPI versus PPI alone for gastroesophageal reflux disease. (**A**): Symptom score change. (**B**): Symptom score after treatment.

**Figure 3 jcm-09-02705-f003:**
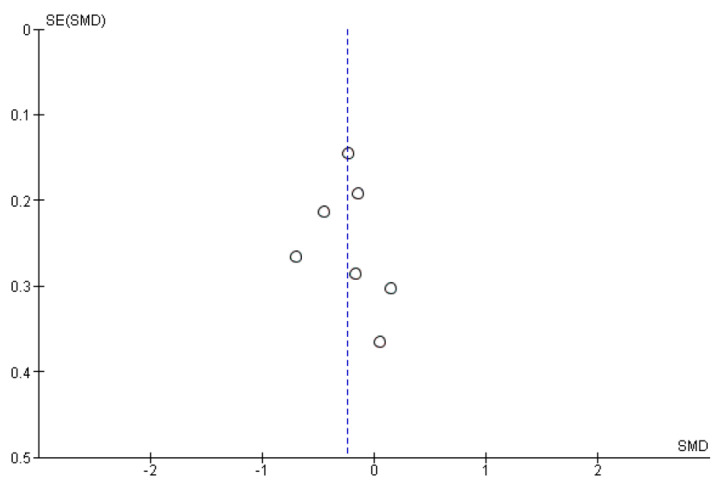
Funnel plot of the included RCTs for symptom score after treatment.

**Figure 4 jcm-09-02705-f004:**
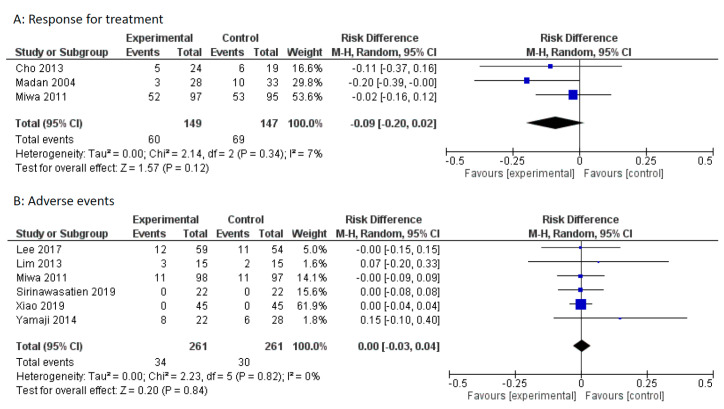
Forest plot of the risk difference (RD) with 95% CI for mosapride combined with PPI versus PPI alone for gastroesophageal reflux disease. (**A**): Response for treatment. Events represent treatment failure. (**B**): Adverse events.

**Figure 5 jcm-09-02705-f005:**
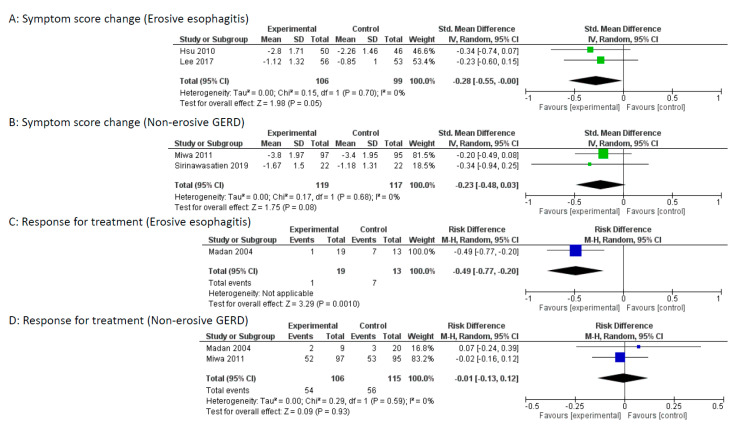
Subgroup analysis between non-erosive gastroesophageal reflux disease (GERD) and erosive esophagitis. (**A**) Forest plot of the standardized mean difference (SMD) with 95% confidence intervals (CI) of symptom score change for mosapride combined with PPI versus PPI alone for erosive esophagitis. (**B**): Forest plot of the SMD with 95% CI of symptom score change for mosapride combined with PPI versus PPI alone for non-erosive GERD. (**C**): Forest plot of the risk difference (RD) with 95% CI of response rate for mosapride combined with PPI versus PPI alone for erosive esophagitis. Events represent treatment failure. (**D**): Forest plot of the RD with 95% CI of response rate for mosapride combined with PPI versus PPI alone for non-erosive GERD. Events represent treatment failure.

**Table 1 jcm-09-02705-t001:** Characteristics of studies included in the systematic review.

Author	Country	Diagnosis of GERD	Assessment of	Range of	Duration	PPI	Allocation	Patients	Age	Gender
	**Year**		**reflux symptoms**	**symptom score**	(weeks)	(dose/day)		**number**	±SD	M/F
Madan	India	GERD symptoms >2/week	Multiplying the scores for severity	0–18	8	Pantoprazole	PPI	33	34.7 ± 10.8	19/14
	2004	Screening endoscopy	(0–3) and frequency (0–3) for 2 items			80 mg	PPI + Mosapride	28	36.5 ± 12.8	21/7
Hsu	Taiwan	GERD symptoms	FSSG; sum of	0–48	4	Lansoprazole	PPI	46	47 ± 8.9	25/21
	2010	Esophagitis at endoscopy	frequency (0–4) for 12 items			30 mg	PPI + Mosapride	50	47 ± 14.8	23/27
Miwa	Japan	GERD symptoms >2/week	Reflux symptoms using 10-cm	0–10	4	Omeprazole	PPI	95	52.2 ± 15.8	35/60
	2011	No esophagitis at endoscopy	visual analogue scale (VAS)			10 mg	PPI + Mosapride	97	52.1 ± 16.1	37/60
Lim	Korea	Typical GERD symptoms	Severity; none (0), mild (1),	0–3	8	Pantoprazole	PPI	15	55.3 (25–66) *	10/5
	2013	Normal gastric emptying scan	moderate (2) or severe (3)			40 mg	PPI + Mosapride	15	48.5 (20–70) *	6/9
Cho	Korea	GERD symptoms >2/week	Severity; none, mild, moderate,	1–5	4	Esomeprazole	PPI	19	43 ± 15	9/10
	2013	Endoscopy, 48 h pH monitoring	severe, or very severe			40 mg	PPI + Mosapride	24	49 ± 16	15/9
Yamaji	Japan	GERD symptoms >2/week	FSSG; sum of	0–48	4	Omeprazole	PPI	28	61.7 ± 11.9	9/19
	2014	Screening endoscopy	frequency (0–4) for 12 items			10 mg	PPI + Mosapride	22	65.0 ± 11.6	4/18
Lee	Korea	GERD symptoms >2/week	Multiplying the scores for severity	0–120	8	Esomeprazole	PPI	53	55.8 ± 8.4	37/16
	2017	Esophagitis at endoscopy	(0–4) and frequency (0–5) for 6 items			40 mg	PPI + Mosapride	56	54.9 ± 11.1	34/22
Sirina	Thailand	GERD symptoms >2/week	FSSG; sum of	0–48	4	Omeprazole	PPI	22	53.1 ± 11.9	6/16
wasatien	2019	Screening endoscopy	frequency (0–4) for 12 items			20 mg	PPI + Mosapride	22	49.2 ± 13.8	7/15
Xiao	China	GERD symptoms and cough	RDQ; sum of frequency (0–5)	0–40	12	Omeprazole	PPI	45	39.9 ± 10.1	29/16
	2019	Screening endoscopy	and severity (0–5) for 4 items			40 mg	PPI + Mosapride	45	40.6 ± 6.0	24/21

GERD: gastroesophageal reflux disease, FSSG: Frequency Scale for Symptoms of GERD, RDQ: Reflux Disease Questionnaire, PPI: proton pump inhibitor, SD: standard deviation, *: range, M/F; male/female.

**Table 2 jcm-09-02705-t002:** Evaluation of bias of RCTs included in the systematic review.

First	Random Sequence	Allocation	Blinding of Participants	Blinding of Outcome	Adequate Assessment	Selective Reporting	No Other
author	generation	concealment	and personnel	assessment	of incomplete outcome	avoided	bias
Madan	Yes	Yes	Yes	Yes	Yes	Yes	Yes
Hsu	Yes	Yes	Yes	Yes	Yes	Yes	Yes
Miwa	Yes	Yes	Yes	Yes	Yes	Yes	Yes
Lim	Yes	Yes	Yes	Yes	Yes	Yes	Yes
Cho	Yes	Yes	Yes	Yes	Yes	Yes	Yes
Yamaji	Yes	Yes	No	No	Yes	Yes	Yes
Lee	Yes	Yes	Yes	Yes	Yes	Yes	Yes
Sirinawasatien	Yes	Yes	Yes	Yes	Yes	Yes	Yes
Xiao	Yes	Unclear	No	No	Yes	Yes	Yes

Yes: Low risk of bias; No: High risk of bias; Unclear: Unclear risk of bias.

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
