# Peer review of "Additional Mosapride to Proton Pump Inhibitor for Gastroesophageal Reflux Disease: A Meta-Analysis"

_jcm, 2020, doi:10.3390/jcm9092705_

Round 1
Reviewer 1 Report
The authors have conducted a meta-analysis of studies comparing a PPI and PPI+mosapride in patients with GERD. The authors have identified 9 studies.
It was found that PPI significantly improves symptom score, but response rate was described in only 3 studies and did not reach significance in the meta-analysis.
Figure 1 is difficult to read and may contain typing or formatting errors (such as age 534 , age 5.3). Please change format.
The Discussion is very short and starts with a one-sentence conclusion with some elaboration of the studied topic later in lines 166 to 172. Several points may be worth mentioned:
- Mosapride may have beneficial effects per se, but may also increase the effect of the PPI. The PPI dose is also of interest as one study used omeprazole 10 mg only, a dose that is so low that it is not in clinical use in all countries. However, the authors are correct in mentioning that the number of studies and subjects is to low to perform sub-group analyses.
- The number of studies is so low that more information about how the GERD diagnosis was made in the nine selected studies should be given, for instance in Table 1.
- Color coded figure illustrating Assessment of methodological quality is missing. This may be the missing Table 2? The table must be submitted for review before the complete manuscript can be considered. Blinding may be considered essential if functional symptoms are studied, which may be the case in many patients with suspected GERD but without effect of PPI.
Author Response
Answer to Reviewer
Thank you for your important comments, which were extremely helpful for improving the quality of our manuscript.
Reviewer #1:
The authors have conducted a meta-analysis of studies comparing a PPI and PPI+mosapride in patients with GERD. The authors have identified 9 studies. It was found that PPI significantly improves symptom score, but response rate was described in only 3 studies and did not reach significance in the meta-analysis. Table 1 is difficult to read and may contain typing or formatting errors (such as age 534 , age 5.3). Please change format.
Thank you very much. The age was corrected, and the format was modified.
The Discussion is very short and starts with a one-sentence conclusion with some elaboration of the studied topic later in lines 166 to 172. Several points may be worth mentioned:
According to your comments, we have enriched the Discussion. We added the discussion about sub-group analysis between non-erosive GERD and erosive esophagitis, cost analysis, and the difference between the previous systematic review (Liu et al.) and out meta-analysis.
Mosapride may have beneficial effects per se, but may also increase the effect of the PPI. The PPI dose is also of interest as one study used omeprazole 10 mg only, a dose that is so low that it is not in clinical use in all countries. However, the authors are correct in mentioning that the number of studies and subjects is to low to perform sub-group analyses. The number of studies is so low that more information about how the GERD diagnosis was made in the nine selected studies should be given, for instance in Table 1.
Color coded figure illustrating Assessment of methodological quality is missing. This may be the missing Table 2? The table must be submitted for review before the complete manuscript can be considered. Blinding may be considered essential if functional symptoms are studied, which may be the case in many patients with suspected GERD but without effect of PPI.
As you mentioned, the difference of PPI dose is an important point. The number of studies and subjects is too low to perform sub-group analyses, and this point was added into the limitation section.
According to your comment, the methods of GERD diagnosis have been added into Table 1.
Table 2 about assessment of methodological quality was on the second page of the Excel file. I'm sorry it was hard to find. We embed the Table 2 in the main manuscript file (Word). In two RCTs, the patients and the outcome assessment were not blinded.

Reviewer 2 Report
The authors performed this study to assess whether mosapride combined with PPIs is better than PPIs alone in improving symptoms of typical GERD. They conclude that the combined treatment is more beneficial than PPis alone. However, there are some features that they should explain more in depth:
- it is not clear how many studies included in their meta-analysis were performed in patients with erosive esophagitis and in patients with non-erosive reflux disease
- The above subdivision is helpful to interpret the results they obtained, because it is well known that PPIs work better in patients with esophagitis than in those without erosions. It is likely that in the former population PPI efficacy does not need to be reinforced by mosapride
- The authors should explain in the discussion of their paper why their study differs from the previous systematic review by Liu et al, who did not find any advantage of the addition of mosapride to PPis in GERD patients
- The combination of PPIs and mosapride implicates a major cost of GERD therapy and this is an important aspect to be discussed
- ref 16 is a systematic review and not a meta-analysis
Author Response
Thank you for your important comments, which were extremely helpful for improving the quality of our manuscript.
Reviewer #2:
The authors performed this study to assess whether mosapride combined with PPIs is better than PPIs alone in improving symptoms of typical GERD. They conclude that the combined treatment is more beneficial than PPis alone. However, there are some features that they should explain more in depth:
it is not clear how many studies included in their meta-analysis were performed in patients with erosive esophagitis and in patients with non-erosive reflux disease
The above subdivision is helpful to interpret the results they obtained, because it is well known that PPIs work better in patients with esophagitis than in those without erosions. It is likely that in the former population PPI efficacy does not need to be reinforced by mosapride
The authors should explain in the discussion of their paper why their study differs from the previous systematic review by Liu et al, who did not find any advantage of the addition of mosapride to PPis in GERD patients
The combination of PPIs and mosapride implicates a major cost of GERD therapy and this is an important aspect to be discussed
ref 16 is a systematic review and not a meta-analysis
Thank you very much. GERD diagnosis and inclusion criteria have been added into Table 1. Two studies (Hsu et al.’s and Lee et al.’s) included patients with just erosive esophagitis, and one study (Miwa et al.’s) included patients with non-erosive reflux disease, four studies included patients with erosive esophagitis or non-erosive reflux disease, one study did not describe the detail of esophageal mucosal break, one study did not performed endoscopy.
Only Madan et al. performed sub-group analysis between non-erosive GERD and erosive esophagitis. Madan et al.’s sub-group analysis, there was no difference in the response rate to either of the regimens (17/20 patients in PPI alone group and 7/9 patients in mosapride combined with PPI group responded; P = 0.63). However, in the erosive esophagitis, more patients who received the combination therapy responded (18/19) than those who received only PPI (6/13), (P = 0.003).
Our meta-analysis additionally performed sub-group analysis between non-erosive GERD and erosive esophagitis. Four studies (Sirinawastien et al.’s, Madan et al.’s, Cho et al.’s and Xiao et al.’s) included patients with erosive esophagitis or non-erosive reflux disease. Among the four studies, the proportions of non-erosive GERD were 86.4%, 47.5%, 58.1%, and 54.4%, respectively.
In Sirinawastien et al.’s study, the majority was non-erosive GERD. Since there was only one study of non-erosive GERD, sub-group analysis between non-erosive GERD and erosive esophagitis has been performed with Sirinawastien et al.’s study as the study of non-erosive GERD.
Subgroup analysis indicated that mosapride combined with a PPI significantly improved the reflux symptom score compared with that of PPI alone for erosive esophagitis group (P = 0.05), but no difference was found for non-erosive GERD group (Figure 5A, 5B). Furthermore, mosapride combined with a PPI significantly enhanced response rate compared with that of PPI alone for erosive esophagitis group (P = 0.001), but no difference was found for non-erosive GERD group (Figure 5C, 5D). The exclusion of Sirinawastien et al.’s study did not change the subgroup analysis results. As you pointed out, it is well known that PPIs work better in patients with esophagitis than in those without erosions. Mosapride may also work better in patients with esophagitis than in those without erosions. These points were added into the revised manuscript (Method section page 6, Result section page 9, Discussion section page 11).
Liu et al. did not find any advantage of the addition of mosapride to PPIs in GERD patients. Although Liu et al.’s study included just 4 RCTs, our study included 9 RCTs. Liu et al.’s analyzed just response rate. On the other hand, we analyzed not only response rate but also symptom score. Response rate was described in only 3 studies and did not reach significance in Liu et al.’s study and our study. However, we have found that mosapride combined with a PPI significantly improved symptom score. These points were added into the Discussion section (page 10).
The cost of esomeprazole is \235 ($2.2) for 40mg (daily dose). The cost of mosapride is \43 ($0.4) for 15mg (daily dose). Mosapride combined with a PPI increases costs by 18%. We consider that this increase in cost would be acceptable for patients receiving PPIs with incomplete symptom control. This point was added into the Discussion section (page 11-12).
The description about Ref 16 was changed to systematic review.

Round 2
Reviewer 2 Report
This reviewer is satisfied by the answers of the authors to his comments.